# BASAHACORPUS: An Expanded Linguistic Resource for Readability Assessment in Central Philippine Languages

**Joseph Marvin Imperial**[Λ,Ω]    **Ekaterina Kochmar**[Υ]
[Λ]University of Bath, UK    [Ω]National University, Philippines
[Υ]MBZUAI, UAE
jmri20@bath.ac.uk  ekaterina.kochmar@mbzuai.ac.ae

## Abstract

Current research on automatic readability assessment (ARA) has focused on improving the performance of models in high-resource languages such as English. In this work, we introduce and release BASAHACORPUS as part of an initiative aimed at expanding available corpora and baseline models for readability assessment in lower resource languages in the Philippines. We compiled a corpus of short fictional narratives written in Hiligaynon, Minasbate, Karay-a, and Rinconada—languages belonging to the Central Philippine family tree subgroup—to train ARA models using surface-level, syllable-pattern, and n-gram overlap features. We also propose a new *hierarchical cross-lingual modeling* approach that takes advantage of a language's placement in the family tree to increase the amount of available training data. Our study yields encouraging results that support previous work showcasing the efficacy of cross-lingual models in low-resource settings, as well as similarities in highly informative linguistic features for mutually intelligible languages.[1]

## 1 Introduction

To ensure optimal reading comprehension, literary resources such as books need to be assigned to readers based on their reading level assessment (Carrell, 1987). Readability assessment can be tackled with a variety of methods ranging from the application of rule-based approaches using such widely accepted formulas as Flesch-Kincaid (Kincaid et al., 1975), to the use of software for linguistic feature extraction such as Coh-Metrix (Graesser et al., 2004) or LFTK (Lee et al., 2021), to the application of extensive machine learning models (Vajjala and Meurers, 2012; Xia et al., 2016). Recently, the latter have been the focus of research on ARA due to the availability of increasingly complex models, including deep learning architectures, that yield

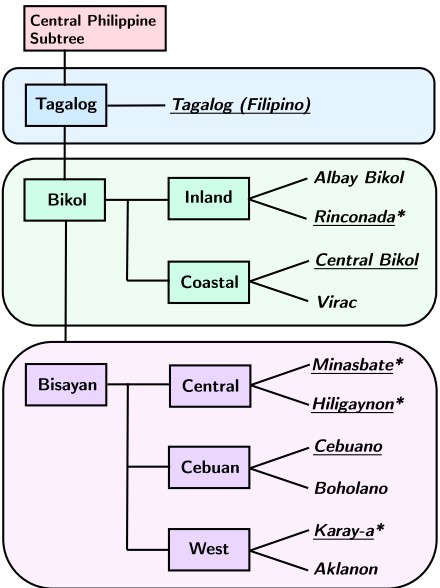

Figure 1: The central subgroup of the Philippine language family tree highlighting the origins of the target languages Hiligaynon, Minasbate, Karay-a, and Rinconada (both underlined and marked with *). Tagalog, Bikol, and Cebuano are also underlined as they are part of further experiments. The complete visualization of this language tree can be found in Zorc (1976).

better text representations and outperform simpler (e.g., formula-based) approaches (Vajjala, 2022). These are, however, mostly practical if trained on high-resource data like English.

Beyond the limelight on high-resource languages like English, languages that do require extensive research efforts and initiatives are commonly considered *low-resource*. One such group of languages comprises over 170 living languages spoken in the Philippines, and one of the three concluding recommendations in the final report of the five-year (2013–2018) USAID Basa Pilipinas ("*Read Philippines*") Program[2] targeting the im-

---

provement of literacy in the Philippines was the *"provision of curriculum-based teaching and learning materials (TLMs) and quality supplementary reading materials"*. This entails the need for a variety of age-appropriate texts for young learners in the Philippines to be accessible at home and in school. Following this, the Department of Education further encouraged the development of reading materials for the Mother Tongue-Based Multilingual Education (MTB-MLE) scheme implemented at primary levels of school education in the Philippines.[3] Thus, the Philippine education sector needs access *not only* to sufficient age-appropriate learning and reading materials *but also* to automated tools for assessing text difficulty that can work effectively for the variety of languages covered by MTB-MLE.

This work contributes a number of resources to the Philippine language research landscape, catering to the challenges identified above, and aligns with other recent research efforts aimed at low-resource languages such as the IndoNLP (Wilie et al., 2020; Winata et al., 2023). First, to increase accessibility and encourage more research endeavors in low-resource Philippine languages, we collect and release BASAHACORPUS,[4] a compilation of children's narratives and short stories from Let's Read Asia online library in four Philippine languages (Hiligaynon, Minasbate, Karay-a, and Rinconda). Second, we train a number of baseline readability assessment models using various cross-lingual setups leveraging the hierarchy in the language family tree. Lastly, we apply a simple model interpretation technique to identify how different linguistic features act as predictors of text complexity for each language.

## 2 A Closer Look into Central Philippine Languages

The Philippines is an archipelago with a rich linguistic background, as evidenced by over 170 languages at the backbone of its culture.[5] In particular, the central Philippine language subtree is considered the largest among other subtrees as it is composed of a number of branches, including the national language *Tagalog* (also known as *Filipino*),

the *Bikol* languages, the *Bisayan* languages, and the *East Mindanao* languages (McFarland, 2004). Focusing on the languages of interest for this study, we show where *Hiligaynon*, *Minasbate*, *Karay-a*, and *Rinconada* are situated in this tree (see Figure 1) and how they are classified within specific subgroups (Zorc, 1976). In the following sections, we integrate the hierarchical relationships of these four languages with other languages in the family tree into our experiments.

### 2.1 Data from Let's Read Asia

Let's Read Asia[6] is an online library of community-translated literary materials sponsored by The Asia Foundation. Using the data collected from the website, we built BASAHACORPUS by compiling short stories written in Philippine languages, which were only available in Hiligaynon, Minasbate, Karay-a, and Rinconada, thus defining our choice to work on these languages. The compiled data are distributed across the first three years of elementary education (L1, L2, and L3). We provide information on the language family, origin subgroup, linguistic vitality, availability of language resources, and whether these languages are used as a medium of instruction in classes in Table 1.

We also compute basic statistical information about the data per language and per level, including mean word and sentence count and vocabulary size, and report it in Table 2. All languages in BASAHACORPUS exclusively use Latin script as is the case for the majority of languages in the Philippines. In terms of the distribution of resources, we obtained explicit permission from Let's Read Asia to use this data in this research with the further goal of publicly sharing the resources, which are licensed under Creative Commons BY 4.0.

## 3 Linguistic Features for Low-Resource Languages

Due to the lack of available NLP tools to extract advanced linguistic information from texts that would work for all four target languages as well as enough data to use deep learning approaches, we derive the same linguistic features as used by the previous work on ARA in other low-resource Philippine languages such as Tagalog (Imperial et al., 2019; Imperial and Ong, 2020, 2021a) and Cebuano (Imperial et al., 2022). However, we note that this study is the first to explore baseline

---

[3]https://www.deped.gov.ph/2016/10/24/mother-tongue-based-learning-makes-lessonsmore-interactive-and-easier-for-students/

[4]*Basaha* generally denotes verbal form of *"read"* in Bisayan languages.

[5]https://www.ethnologue.com/country/PH/

[6]https://www.letsreadasia.org/

| Language | Family | Indigenous | Vitality | Instruction | Digital Language Support |
|---|---|---|---|---|---|
| Hiligaynon (HIL) | Central | No | Insitutional | Yes | Ascending |
| Minasbate (MSB) | Central | No | Insitutional | As a subject | Still |
| Kinaray-a (KRJ) | West | Stable indigenous | Insitutional | Yes | Emerging |
| Rinconada (BTO) | Inland Bikol | Stable indigenous | Stable | As a subject | Still |

Table 1: Description of the language data for Hiligaynon, Minasbate, Karay-a, and Rinconada. We also include important information from Ethnologue about each language's classification and status, including origin subgroup (identified as indigenous or not), linguistic vitality, whether a language is used as a medium of instruction or as a subject, and the availability of online digital resources (for terminology, please refer to the Ethnologue page).

| Language | Total | Level | Document Count | Mean Word Count | Mean Sent Count | Vocabulary |
|---|---|---|---|---|---|---|
| Hiligaynon (HIL) | 133 | L1 | 65 | 198.8 | 20.7 | 2043 |
| | | L2 | 22 | 296.5 | 39.1 | 1539 |
| | | L3 | 46 | 610.0 | 57.3 | 4137 |
| Minasbate (MSB) | 271 | L1 | 124 | 240.4 | 28.7 | 3836 |
| | | L2 | 77 | 360.5 | 43.0 | 4097 |
| | | L3 | 70 | 578.6 | 62.5 | 5520 |
| Karay-a (KRJ) | 177 | L1 | 61 | 191.0 | 21.4 | 1937 |
| | | L2 | 34 | 410.9 | 47.7 | 2309 |
| | | L3 | 82 | 569.1 | 60.3 | 6264 |
| Rinconada (BTO) | 195 | L1 | 117 | 261.0 | 30.0 | 4222 |
| | | L2 | 36 | 521.7 | 59.5 | 3313 |
| | | L3 | 42 | 505.2 | 55.1 | 3958 |

Table 2: Overview and basic statistics of the data included in the BASAHACORPUS per language and per grade level with the total number of documents. The vocabulary denotes the total count of unique words per class and per language. An increasing linear relationship can be observed for both mean word count and mean sentence count per document as the level increases.

modeling of text complexity in the four selected languages – Hiligaynon, Minasbate, Karay-a, and Rinconada. We list all extracted features below.

**Traditional Features**. We extract 7 frequency-based features using count-based text characteristics. Despite being often considered simplistic, these predictors have consistently proven to be effective in text readability detection (Pitler and Nenkova, 2008; Imperial and Ong, 2021b). As in the previous work on a Cebuano dataset Imperial et al. (2022), our extracted text-level features include the *total number of unique words, total number of words per document, average word length per document, average number of syllables (per word), total sentence count per document, average sentence length per document*, and the *total number of polysyllable words*.

**Syllable-pattern Orthographic Features**. We also extract 11 orthographic predictors taking into account all possible syllable patterns in the Philippine language space. These consonant

($c$) and vowel ($v$) patterns include *v, cv, vc, cvc, vcc, ccv, cvcc, ccvcc, ccvccc*. We also extract *consonant clusters* and measure the average length of consonant groups in a word without an intervening vowel which have been proven to influence reading difficulty (Chard and Osborn, 1999).

**Cross-lingual N-gram Overlap**. A new CROSS-NGO feature has recently been introduced by Imperial and Kochmar (2023), which exploits mutual intelligibility or the degree of language relatedness via n-gram overlap between languages from the same family tree. Since the study reported a significant boost in performance for readability assessment in low-resource languages using this feature, we see a clear practical application for it in this work. We estimate the *bigram* and *trigram overlap* between the four target central Philippine languages (Hiligaynon, Minasbate, Karay-a, and Rinconada), as well as the three parent languages (Tagalog, Bikol, and Cebuano) for a total of 14 new features. As an additional reference, we add two

tables quantifying the similarities via cross-lingual n-gram overlap of the 7 Philippine languages used in this study in Tables 7 and 6 of Appendix A.

## 4 Hierarchical Cross-lingual Modeling

Cross-lingual modeling for automatic readability assessment typically involves extracting language-independent features or experimenting with a classifier trained on the data in one language and applied to comparable data in another language (Madrazo Azpiazu and Pera, 2020; Weiss et al., 2021; Lee and Vajjala, 2022; Imperial and Kochmar, 2023). In this study, we introduce a new approach to training readability assessment models using close relationships between languages in the family tree, which we refer to as **hierarchy-based cross-lingual modeling**. For this setup, our cross-lingual training process is divided into iterations involving different combinations of language data in the training split with respect to the hierarchy the languages occupy in the Central Philippine language subtree, as illustrated in Figure 1. We list all such feature combinations below:

**Monolingual (L).** This setup uses the standard train-test split involving one language only.

**Lang + Parent Lang (L+P).** This setup combines the parent language or *lingua franca* with respect to their subgroup. For Hiligaynon, Minasbate, and Karay-a, belonging to the Bisayan languages in the Visayas region, we add the Cebuano language, while for Rinconada, classified as a member of the Bikol languages belonging to the Luzon region, we add the Central Bikol language both from Imperial and Kochmar (2023).

**Lang + National Lang (L+N).** This setup combines the four target languages with Tagalog in the training data extracted from Imperial and Kochmar (2023), Imperial and Ong (2020) and Imperial et al. (2019). Tagalog is the recognized national language taught as a subject in all elementary and secondary schools in the Philippines.

**Lang + Parent Lang + National Lang (L+P+N).** This setup combines each target language with its corresponding parent language and Tagalog in the training data.

**All Langs (*L).** This setup combines all seven languages used in the previous iterations (Hiligaynon, Minasbate, Karay-a, Rinconada, Bikol, Cebuano, and Tagalog) in the training data.

We note that the Tagalog, Bikol, and Cebuano data from Imperial and Kochmar (2023) that we use for this study also follows the same category distribution (L1, L2, and L3) as our four languages from Table 2. For all model training in this study, we use Random Forest with default hyperparameters from Scikit-Learn (Pedregosa et al., 2011), as the efficacy of this algorithm was reported in the previous studies on ARA in the Philippine languages (Imperial et al., 2022; Imperial and Ong, 2021a). We perform stratified 5-fold sampling for all three classes to be properly represented. Hyperparameter values of the Random Forest model can also be found in Table 5 of Appendix A.

## 5 Results and Discussion

Below, we summarize and discuss the most notable insights obtained from the conducted experiments.

| Language | L | L+P | L+N | L+P+N | *L |
|---|---|---|---|---|---|
| **HIL** | 0.697 | 0.666 | 0.629 | 0.592 | **0.814** |
| **MSB** | 0.641 | 0.555 | 0.611 | 0.574 | **0.648** |
| **KRJ** | 0.618 | 0.628 | 0.657 | 0.629 | **0.685** |
| **BTO** | 0.682 | **0.846** | 0.743 | 0.714 | 0.717 |

Table 3: Accuracy across the experiments described in Section 4 using various combinations of language datasets and leveraging hierarchical relations between languages.

### 5.1 Using extensive multilingual training data results in generally better performance.

Table 3 shows the results of the ablation experiments using the hierarchical cross-lingual modeling setup as described in Section 4. We observe that using the all languages setup (*L) for the training data helps obtain the best accuracy performance for the three Bisayan languages (HIL, MSB and KRJ) but not for Rinconada (BTO) from the Bikol subgroup. However, the score for Rinconda with the *L setup is still higher than for the monolingual model and even the L+P+N setup. A $t$-test confirms that the scores obtained with the *L setup are significantly higher than those obtained with the monolingual setup L for all languages at $\alpha = 0.05$ level ($p = 0.048$). These results support the findings of Imperial and Kochmar (2023), who showed the importance of combining data from closely related

languages (or languages within the same family tree) for performance improvements in ARA models.

## 5.2 Stronger mutual intelligibility improves model performance.

Following Imperial and Kochmar (2023), we compute the overlap of the top 25% of the trigrams and bigrams in order to estimate mutual intelligibility between languages from the Bisayan and Bikol subgroups and their respective parent languages, Cebuano and Bikol. We find that Rinconada (BTO) has the highest overlap (0.696 for trigrams and 0.887 for bigrams) with its parent language (Bikol) – a fact that explains why the best results for this language are obtained with the L+P combination. For comparison, the other three languages (Hiligaynon, Minasbate, and Karay-a) show $n$-gram overlaps of 0.609, 0.579, and 0.540 for trigrams, and 0.863, 0.789, and 0.809 for bigrams with their parent language, respectively. See the trigram and bigram overlap results in Tables 6 and 7 in Appendix A.

## 5.3 Support for traditional features in approximating text complexity for Philippine languages.

Finally, we identify the most informative linguistic features by calculating the mean decrease in impurities when splitting by the Random Forest models trained with all languages (*L) and applied to the test splits for each of the four languages. Table 4 shows that all models for all languages demonstrate the same order in top features which are all considered count-based predictors. We believe that this finding corroborates results from previous work showing that frequency-based features such as word and sentence counts are still viable measures of text complexity for Philippine languages (Macahilig, 2014; Gutierrez, 2015).

## 6 Related Work

In the past, cross-lingual modeling was applied to classification-based ARA both for non-related (Vajjala and Rama, 2018; Madrazo Azpiazu and Pera, 2020; Weiss et al., 2021; Lee and Vajjala, 2022; Mollanorozy et al., 2023) and highly-related languages (Imperial and Kochmar, 2023), with the latter reporting favorable results under predefined language similarity constraints such as high n-gram overlap.[7] Research efforts that greatly contribute to

---

[7]Here, we interpret *language relatedness* as a measure of similarity of linguistic characteristics such as n-gram overlap

| HIL | | MSB | |
|---|---|---|---|
| *word count* | 0.096 | *word count* | 0.111 |
| *sentence count* | 0.079 | *sentence count* | 0.077 |
| *polysyll count* | 0.053 | *polysyll count* | 0.049 |
| *avg sent len* | 0.045 | *avg sent len* | 0.039 |
| *tag trigram sim* | 0.038 | *tag trigram sim* | 0.037 |

| KRJ | | BTO | |
|---|---|---|---|
| *word count* | 0.102 | *word count* | 0.115 |
| *sentence count* | 0.074 | *sentence count* | 0.072 |
| *polysyll count* | 0.052 | *polysyll count* | 0.047 |
| *avg sent len* | 0.042 | *avg sent len* | 0.044 |
| *tag trigram sim* | 0.036 | *tag trigram sim* | 0.037 |

Table 4: Most informative features per language identified using mean decrease in impurity scores calculated for the Random Forest models.

the development of low-resource and cross-lingual readability assessment systems often focus on corpus building and the development of baseline models. Previous efforts of this kind have covered a wide array of languages, including Bangla (Islam et al., 2012; Islam and Rahman, 2014), Tibetan (Wang et al., 2019), Arabic (Saddiki et al., 2018; Al Khalil et al., 2020), Vietnamese (Doan et al., 2022), Bengali (Chakraborty et al., 2021), and Bikol (Imperial and Kochmar, 2023). As compared to the previous works, ours is the first study in readability assessment that investigates the effects of different language hierarchies in modeling text complexity for languages belonging to different subgroups of a greater family tree, with the focus on central Philippine languages.

## 7 Summary

We introduce BASAHACORPUS, a compilation of language resources that includes a collected corpus of short stories and baseline readability assessment models for four central Philippine languages. We show that, through a *hierarchical cross-lingual modeling approach*, a readability model trained with all the available Philippine language data generally performs better compared to using single-language datasets. Through model interpretation, we also provide further support for the use of frequency-based features such as word and sentence counts as effective predictors of complexity in Philippine languages. This study serves as a response to the call for more research efforts, theoretically grounded baselines, and accessible data for low-resource languages.

---

(Imperial and Kochmar, 2023).

## Limitations

**Limited feature sets for low-resource languages.** Due to severely limited existing NLP research efforts for the Philippine languages that this work addresses, we have to resort to a small number of feature extraction methods covering surface-level characteristics, syllable patterns, and n-gram overlap that have been previously applied to the related Philippine languages such as Tagalog and Cebuano (Imperial and Ong, 2020, 2021a; Imperial et al., 2022; Imperial and Kochmar, 2023). Nevertheless, we believe that our findings are valuable as this work provides the baseline for readability assessment in Hiligaynon, Minasbate, Karay-a, and Rinconada in addition to being the first one to address these languages. We hope that future research efforts will lead to a substantial expansion in the data available for these languages, which in turn will help researchers develop and apply more advanced ARA models to these languages and benchmark them against the results reported in our paper. We consider our work the first step in the direction of addressing such challenging tasks as ARA in low-resource Philippine languages.

**Low variety of the data.** This work uses only fictional short stories in specific grade levels (L1, L2, and L3) as training data for the readability assessment models, which may be considered a limitation in the application domain. While the same features can be extracted and applied to other text forms in various domains such as news articles or poems, we do not claim that the results will generalize or apply to such other datasets.

## Ethics Statement

We foresee no serious ethical implications from this study.

## Acknowledgements

We thank the anonymous reviewers for their constructive feedback and the ACs, SACs, and PCs for their appreciation of this work. We also thank the community translators and maintainers of the online library of Let's Read Asia for keeping the digital resources in the Philippine languages freely available for everyone. JMI is supported by the UKRI Centre for Doctoral Training in Accountable, Responsible, and Transparent AI (ART-AI) [EP/S023437/1] of the University of Bath and by the Study Grant Program of National University Philippines.

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

# A   Appendix

| Hyperparameter | Value |
|---|---|
| n_estimators | 100 |
| criterion | gini |
| max_depth | None |
| min_samples | 1 |
| max_features | sqrt |
| random_state | none |

Table 5: Hyperparameter settings for the Random Forest algorithm used for training the models with scikit-learn. These are also default values preset in WEKA 0.2.

| | HIL | MSB | KRJ | BTO | TGL | CEB | BCL |
|---|---|---|---|---|---|---|---|
| **HIL** | 1.000 | 0.713 | 0.741 | 0.610 | 0.590 | 0.609 | 0.604 |
| **MSB** | 0.713 | 1.000 | 0.696 | 0.673 | 0.586 | 0.579 | 0.649 |
| **KRJ** | 0.741 | 0.696 | 1.000 | 0.622 | 0.552 | 0.540 | 0.593 |
| **BTO** | 0.610 | 0.673 | 0.622 | 1.000 | 0.615 | 0.528 | 0.696 |
| **TGL** | 0.590 | 0.586 | 0.552 | 0.615 | 1.000 | 0.628 | 0.588 |
| **CEB** | 0.609 | 0.579 | 0.540 | 0.528 | 0.628 | 1.000 | 0.533 |
| **BCL** | 0.604 | 0.649 | 0.593 | 0.696 | 0.588 | 0.533 | 1.000 |

Table 6: Trigram overlap between all seven Central Philippine languages.

| | HIL | MSB | KRJ | BTO | TGL | CEB | BCL |
|---|---|---|---|---|---|---|---|
| **HIL** | 1.000 | 0.830 | 0.859 | 0.831 | 0.834 | 0.863 | 0.818 |
| **MSB** | 0.830 | 1.000 | 0.831 | 0.863 | 0.782 | 0.789 | 0.847 |
| **KRJ** | 0.859 | 0.831 | 1.000 | 0.811 | 0.796 | 0.809 | 0.816 |
| **BTO** | 0.831 | 0.863 | 0.811 | 1.000 | 0.786 | 0.785 | 0.887 |
| **TGL** | 0.834 | 0.782 | 0.796 | 0.786 | 1.000 | 0.812 | 0.810 |
| **CEB** | 0.863 | 0.789 | 0.809 | 0.785 | 0.812 | 1.000 | 0.789 |
| **BCL** | 0.818 | 0.847 | 0.816 | 0.887 | 0.810 | 0.789 | 1.000 |

Table 7: Bigram overlap between all seven Central Philippine languages.