# OpenReview forum: "BasahaCorpus: An Expanded Linguistic Resource for Readability Assessment in Central Philippine Languages"
_EMNLP/2023/Conference — EMNLP 2023 Main_

### Official Review · Reviewer_PKp6 · 2023-08-04

**Soundness:** 4

**Excitement:**

3: Ambivalent: It has merits (e.g., it reports state-of-the-art results, the idea is nice), but there are key weaknesses (e.g., it describes incremental work), and it can significantly benefit from another round of revision. However, I won't object to accepting it if my co-reviewers champion it.

**Paper Topic And Main Contributions:**

This paper proposed a new dataset for readability assessment consisting of four low-resource Philippine languages. It trained random forest models on the different combinations of these languages and showed the importance of combining data from closely related languages.

**Reasons To Accept:**

The introduction of low-resource Philippine languages and education shows strong motivation for this research, and hence the development of the corpus is an excellent contribution to the academic area.

**Reasons To Reject:**

The main weakness of this paper is that it did not mention why it introduced a new method. According to Lines 171-186, we can see that previous works have already used different languages to enhance the models. However, the paper did not compare the proposed method with previous works or show their differences.

**Reproducibility:**

5: Could easily reproduce the results.

**Reviewer Confidence:**

4: Quite sure. I tried to check the important points carefully. It's unlikely, though conceivable, that I missed something that should affect my ratings.

---

> ### Author Rebuttal · Authors · 2023-08-28
>
> We are extremely grateful for the reviewer’s feedback on our work and very much appreciate the positive comments regarding our contributions to the corpora building for readability assessment in Philippine languages. Please find our responses to the individual questions below. We hope that our responses address both the questions and the weaknesses identified, and we also hope that this may be reflected in the updated scores.
>
> 1. *"The main weakness of this paper is that it did not mention why it introduced a new method. The paper did not compare the proposed method with previous works or show their differences."* - Our paper is inspired by the previous work of Imperial and Kochmar (2023), which describes the state-of-the-art approach in cross-lingual ARA in the Philippine languages. As a short paper, it focuses on an extension of this approach, and our original contribution to this field is the exploration of the hierarchical relations between low-resource languages as is demonstrated in Figure 1–thus, we call our approach “hierarchy-based cross-lingual ARA”. Table 2 shows the direct comparison and advantages of using our approach for ARA in low-resource languages. We thank the reviewer for making this point, and we will revise our paper accordingly (specifically, the discussion in Section 2) to emphasize further the motivation and theoretical background of hierarchical cross-lingual ARA in application to low-resource languages – Philippine languages in this case.

---

### Official Review · Reviewer_KDgi · 2023-08-05

**Soundness:** 3

**Excitement:**

3: Ambivalent: It has merits (e.g., it reports state-of-the-art results, the idea is nice), but there are key weaknesses (e.g., it describes incremental work), and it can significantly benefit from another round of revision. However, I won't object to accepting it if my co-reviewers champion it.

**Missing References:**

The paper does cite some relevant references on Readability Assessment- however it does not summarize this past work in relation to the current contributions. Further, there are few references on ARA on low resource languages in general (apart from the Central Phillipine languages)

**Paper Topic And Main Contributions:**

This paper describes a new Readability assessment corpus for 4 Central Phillipine languages. These are low-resource languages.
Further, it demonstrates baseline results for ARA (automatic readability assessment) using typological information. The authors mention that code and data will be released upon publication using a Creative Commons License

**Questions For The Authors:**

A) Did the authors specifically choose Random Forest as a model due to small data size? What about other models?
B) If the data is in Latin script, then to what extent does it capture the sound(phonemic) differences between the languages? This has a bearing on how effectively the typological information is really being used

**Reasons To Accept:**

The work describes a resource for a  group of languages that are not well represented in NLP. It also situates the need for this work within the context of education in a multi-lingual country.

**Reasons To Reject:**

As the authors are primarily releasing the Basaha Corpus as a resource, the paper contains very few details about the corpus itself. E.g. Table 1 in the corpus describes the 4 languages and their status as per Ethnologue. But the numbers under L1 L2 and L3 are not explained- are these the number of articles/words/sentences? Some clear indication of the size of the corpus would have been more insightful here. Further, what is the script used for these languages - is it Latin? Can this affect feature extraction in terms of syllables for these languages? These points are not made clear in the paper.
The paper extensively cites previous work on Tagalog and Cebuano. A brief summary of this previous work would have been helpful to evaluate against the contribution made by this paper.
The cross-lingual modeling approach showed promising results, but again a little more in depth characterization of these results would have helped. To what extent is the data size affecting the results? It's hard to say given some missing details in the paper.

**Reproducibility:**

3: Could reproduce the results with some difficulty. The settings of parameters are underspecified or subjectively determined; the training/evaluation data are not widely available.

**Reviewer Confidence:**

4: Quite sure. I tried to check the important points carefully. It's unlikely, though conceivable, that I missed something that should affect my ratings.

---

> ### Author Rebuttal · Authors · 2023-08-28
>
> We are extremely grateful for the reviewer’s feedback on our work and very much appreciate the positive comments regarding our contributions to the corpora building for readability assessment in Philippine languages. Please find our responses to the individual questions below. We hope that our responses address both the questions and the weaknesses identified, and we also hope that this may be reflected in the updated scores.
>
> 1. *"The numbers under L1 L2 and L3 are not explained - are these the number of articles/words/sentences?"* - We apologize for not being clear enough with the details on the data presented in the tables. These values denote the number of documents categorized as belonging to each level.
>
> 2. *"The cross-lingual modeling approach showed promising results, but again, a little more in-depth characterization of these results would have helped. To what extent is the data size affecting the results?"*  - We thank the reviewer for raising this point. We believe that the question on the data size effects deserves further investigation, however, we are limited in our ability to fully investigate this aspect: given that we are working with low-resource languages, we can only report the results obtained with the available amount of data. We believe that if more data were available for these languages, we would be able to show further improvements on the current results (e.g., although not directly comparable, the results obtained on such high-resource languages as English using similar feature-based systems are higher than what we show in our work).
>
> 3. *"Did the authors specifically choose Random Forest as a model due to the small data size? What about other models?"* - The choice of the Random Forest model is motivated by the extensive previous work on the Philippine languages by Imperial and Ong (2019-2021), which explored other algorithms such as Logistic Regression, SVM, and Random Forest for readability assessment and showed that the Random Forest model achieved the best results and demonstrated higher robustness to misclassifications. Thus, we used this algorithm to train the baseline models in our work.
>
> 4. *"Is the data in Latin script?"* - Yes, all major languages in the Philippines, including the four used in the study, use the Latin alphabet.
>
> 5. *"Can this affect feature extraction in terms of syllables for these languages?"* - Our scripts for syllable pattern features are only applicable to languages using the Latin alphabet and will not be applicable to languages using a different writing system. We thank the reviewer for pointing this out, and we will specifically include a discussion on this in our Limitations section.
>
> 6. *"To what extent does it capture the sound(phonemic) differences between the languages?"* - Currently, our study is only limited to text-based features and does not cover phonemic or sound-based patterns due to the limited availability of NLP tools to extract more advanced features. We thank the reviewer for suggesting this possible future research direction and will be adding this to the Limitations section.

---

### Official Review · Reviewer_v7Sy · 2023-08-05

**Soundness:** 4
**Typos Grammar Style And Presentation Improvements:** The paper is generally well structure…

**Excitement:**

3: Ambivalent: It has merits (e.g., it reports state-of-the-art results, the idea is nice), but there are key weaknesses (e.g., it describes incremental work), and it can significantly benefit from another round of revision. However, I won't object to accepting it if my co-reviewers champion it.

**Paper Topic And Main Contributions:**

The paper presents a dataset for the assessment of readability for languages that belong to the central Philippine subgroup. The languages covered have not been treated before. The paper briefly presents the features taken into account, the experimentation setting and baseline developed leading to promising results. The author(s) make use of traditional widely used linguistic features plus the previously introduced cross-lingual N-gram overlap feature and introduce the hierarchical cross-lingual modeling for ARA.

The main contributions of the paper can be defined as follows:
- a new corpus for automatic readability assessment for low-resourced languages;
- the baseline developed;
- the results support previous work for other languages;
- the dataset will be freely available upon acceptance.

**Questions For The Authors:**

It is mentioned in the paper: The compiled data are distributed across the first three years of elementary education (L1, L2 and L3) (l 109 - 111). : how are levels assigned to the texts? What is the difference between i.e.,  L2 and L3 levels? What is the confidence for the manual distribution of texts to levels?

The work presented here is focused on L1, L2 and L3 levels. Isn't that a limitation to be clearly stated in the respective section?
(it is implied now).

Table 7: What does Vocab stand for?



**Reasons To Accept:**

The main strengths of the paper can be summarised as follows:
- the task is interesting and timely;
- the dataset presented is intended for a group of less-resourced languages that have not been treated before;
- the paper introduces the notion of hierarchical cross-lingual modeling;
- the results obtained support the findings of previous work applied to new languages with mutual intelligibility.


**Reasons To Reject:**

The main weaknesses of the paper can be summarised as follows:
- the research is limited to a L1, L2, and L3 levels (also reported in the limitations section);
- the paper includes a long introduction to the task and background information, but L1, L2, and L3 levels are not explained;
- the paper does not provide meaningful comparisons with previous work;
- the results obtained are mentioned to support the findings of previous work - yet the


**Reproducibility:**

3: Could reproduce the results with some difficulty. The settings of parameters are underspecified or subjectively determined; the training/evaluation data are not widely available.

**Reviewer Confidence:**

4: Quite sure. I tried to check the important points carefully. It's unlikely, though conceivable, that I missed something that should affect my ratings.

---

> ### Author Rebuttal · Authors · 2023-08-28
>
> We are extremely grateful for the reviewer’s feedback on our work and very much appreciate the positive comments regarding our contributions to the corpora building for readability assessment in Philippine languages. Please find our responses to the individual questions below. We hope that our responses address both the questions and the weaknesses identified, and we also hope that this may be reflected in the updated scores.
>
> 1. *"The paper does not provide meaningful comparisons with previous work; the results obtained are mentioned to support the findings of previous work."* - Since this is a short paper, due to space limitations, we did not include a separate section on related works. However, we made sure to cite and refer to the results that previous research has reported throughout our paper — in particular, in Sections 3, 4, and 5. We cited previous works to support their findings as reflected by our results using our new proposed method of hierarchy-based cross-lingual ARA. If our paper is accepted, we will be happy to use extra space for a dedicated section on related work.
>
>
> 2. *"How are levels assigned to the texts? What is the difference between i.e., L2 and L3 levels? What is the confidence for the manual distribution of texts to levels?"* - The levels are assigned by experts from the Let’s Read Asia group. As mentioned in Section 2.1, each level is used in the context of the Philippine educational system. For example, L1 is used for first graders, L2 is for second graders, and so on. We extracted all available data for each level and language from Let’s Read Asia, and this distribution is reflected in Table 7.
>
>
> 3. *"The research is limited to L1, L2, and L3 levels."* - We acknowledge that this is a limitation of our study, however, we contend that similar limitations apply to all low-resource language studies. To clarify, there are seven grade levels in the Philippine educational system in total, but our multilingual corpus only covers the first three levels due to limited data availability from the source itself (Let’s Read Asia). However, as explained in the Introduction, machine-readable data in these languages are not yet available to the NLP community. Thus, the release of this newly compiled baseline corpus and the baseline readability models trained on it may serve as an important stepping stone to encourage more development and research on these low-resource Philippine languages in the future. Taking your feedback into account, we will include a clear definition of the three levels and add a separate paragraph in the Limitations section discussing these points in the revised version of the paper.
>
>
> 4. *"What does Vocab stand for?"* - Vocab (or vocabulary) is the number of unique terms per class in a corpus. We will include this definition in the revised version of the paper.

---

### Meta-Review · Area_Chair_HKvK · 2023-09-24

**Recommendation:** 4

**Metareview:**

Summary (from reviewer v7Sy): The paper presents a dataset for the assessment of readability for languages that belong to the central Philippine subgroup. The languages covered have not been treated before. The paper briefly presents the features taken into account, the experimentation setting and baseline developed leading to promising results. The author(s) make use of traditional widely used linguistic features plus the previously introduced cross-lingual N-gram overlap feature and introduce the hierarchical cross-lingual modeling for automatic readability assessment (ARA).

The reviews acknowledged the value of the resource that was developed for this paper, and its unique position in both the languages it covers and the domain (ARA). Many of the indigenous Philippine languages lack significant text resources altogether, let alone resources in ARA.

The discussion period was very productive. While the reviewers pointed out a number of missing or not understood statistics and other pieces of information regarding the corpus, there was also some confusion about why the specific modeling approach was used. The authors’ responses helped clear all of this up to the point where any revisions required to the paper represent missing or not fully explained information, but not methodological flaws or soundness issues that should lead to rejection. However, the authors must carefully address all questions raised in revisions if the paper is accepted.

It is worth noting that this is a paper that primarily provides a resource, and does so for a set of less-resourced languages. Papers of this type often receive medium or low enthusiasm scores, despite the significant positive impact that publishing them in popular venues can have.

---

### Decision · Program_Chairs · 2023-10-07

**Decision:**

Accept-Main

**Comment:**

Summary (from reviewer v7Sy): The paper presents a dataset for the assessment of readability for languages that belong to the central Philippine subgroup. The languages covered have not been treated before. The paper briefly presents the features taken into account, the experimentation setting and baseline developed leading to promising results. The author(s) make use of traditional widely used linguistic features plus the previously introduced cross-lingual N-gram overlap feature and introduce the hierarchical cross-lingual modeling for automatic readability assessment (ARA).

The reviews acknowledged the value of the resource that was developed for this paper, and its unique position in both the languages it covers and the domain (ARA). Many of the indigenous Philippine languages lack significant text resources altogether, let alone resources in ARA.

The discussion period was very productive. While the reviewers pointed out a number of missing or not understood statistics and other pieces of information regarding the corpus, there was also some confusion about why the specific modeling approach was used. The authors’ responses helped clear all of this up to the point where any revisions required to the paper represent missing or not fully explained information, but not methodological flaws or soundness issues that should lead to rejection. However, the authors must carefully address all questions raised in revisions if the paper is accepted.

It is worth noting that this is a paper that primarily provides a resource, and does so for a set of less-resourced languages. Papers of this type often receive medium or low enthusiasm scores, despite the significant positive impact that publishing them in popular venues can have.